# RNA Dynamics in Alzheimer’s Disease

**DOI:** 10.3390/molecules26175113

**Published:** 2021-08-24

**Authors:** Agnieszka Rybak-Wolf, Mireya Plass

**Affiliations:** 1Max Delbrück Center for Molecular Medicine (MDC), Berlin Institute for Medical Systems Biology (BIMSB), 10115 Berlin, Germany; 2Gene Regulation of Cell Identity, Regenerative Medicine Program, Bellvitge Institute for Biomedical Research (IDIBELL), L’Hospitalet del Llobregat, 08908 Barcelona, Spain; 3Program for Advancing Clinical Translation of Regenerative Medicine of Catalonia, P-CMR[C], L’Hospitalet del Llobregat, 08908 Barcelona, Spain; 4Center for Networked Biomedical Research on Bioengineering, Biomaterials and Nanomedicine (CIBER-BBN), 28029 Madrid, Spain

**Keywords:** Alzheimer’s disease, neurodegenerative diseases, RNA binding proteins, RNA processing, post-transcriptional regulation, alternative splicing, miRNAs, circRNAs, lncRNAs

## Abstract

Alzheimer’s disease (AD) is the most common age-related neurodegenerative disorder that heavily burdens healthcare systems worldwide. There is a significant requirement to understand the still unknown molecular mechanisms underlying AD. Current evidence shows that two of the major features of AD are transcriptome dysregulation and altered function of RNA binding proteins (RBPs), both of which lead to changes in the expression of different RNA species, including microRNAs (miRNAs), circular RNAs (circRNAs), long non-coding RNAs (lncRNAs), and messenger RNAs (mRNAs). In this review, we will conduct a comprehensive overview of how RNA dynamics are altered in AD and how this leads to the differential expression of both short and long RNA species. We will describe how RBP expression and function are altered in AD and how this impacts the expression of different RNA species. Furthermore, we will also show how changes in the abundance of specific RNA species are linked to the pathology of AD.

## 1. Introduction

Alzheimer’s disease (AD) is the most common type of age-related dementia for which no cure is yet available. Currently, it is estimated that 10–30% of the population older than 65 years has AD [1], from which only about 1–5% of the cases have clearly known genetic causes [2].

One of the main characteristics of AD is that it is a slowly progressing neurodegenerative disease. It has been shown that AD-associated intraneural lesions can occur in young individuals [3,4] several decades before the appearance of cognitive symptoms, which is known as the preclinical phase of AD [5]. This phase is later followed by the appearance of cognitive symptoms that begin with mild memory loss that gradually turns towards a severe impairment of executive and cognitive functions over the course of one or two decades [6].

At the molecular level, AD is characterized by the accumulation of amyloid β (Aβ) peptides in plaques and the presence of phosphorylated Tau aggregates in neurofibrillary tangles (NFT) [7]. For decades, it was thought that AD-related cognitive impairments were caused by the accumulation of a Aβ peptides in the brain parenchyma. The so-called amyloid hypothesis suggested that the formation of amyloid plaques that lead to neuronal degeneration and death was the cause of AD [8]. This hypothesis has been the lead target for drug development to treat AD. However, clinical trials targeting Aβ remain unsuccessful in improving cognitive function or slowing down disease progression [9]. Therefore, these results suggest that the accumulation of Aβ peptides, as posed in the amyloid hypothesis, cannot itself explain the onset and progression of AD.

Recently, a new focus has been placed on the role of RNA processing and its impact on neurodegenerative diseases. RNA binding proteins (RBPs) are the main regulators of gene regulation at the RNA level, including transcription, processing, transport, and degradation. In this review, we will summarize the current evidence showing how RBP expression and function are altered in AD and how these changes affect the biogenesis, expression, processing, and localization of coding and non-coding RNAs (ncRNAs) in AD, as well as their impact on AD pathology.

## 2. RNA Processing Governs the RNA Makeup of Cells

Within an organism, each cell expresses a different combination of genes that constitutes its unique fingerprint. This fingerprint, known as the transcriptome, reflects several aspects of the cell, allowing us to identify, among others, its function, its activation state, and even its response to the surrounding environment [10,11,12]. Several interconnected mechanisms and processes are involved in controlling the specific RNA makeup of a cell, including transcription, splicing, polyadenylation, nuclear export, transport, and turnover [13]. Apart from transcription, which requires the recognition of specific signals in the DNA, all mechanisms that regulate gene expression post-transcriptionally are largely regulated by RNA binding proteins (RBPs).

RBPs are proteins that bind to RNA and regulate their fate and function post-transcriptionally. RBPs function both in the nucleus and the cytoplasm, although their specific roles are compartment dependent. In the nucleus, RBPs mainly regulate RNA processing, including splicing, polyadenylation, and export. In the cytoplasm, RBPs regulate RNA silencing, degradation, transport, and translation, as well as both protein and RNA localization (Figure 1).

RBPs have essential functions in the brain, and they are involved in the regulation of key processes such as neurogenesis, synaptic transmission, and plasticity [14]. Altered function of RBPs is commonly observed in neurodegenerative diseases [14,15,16,17,18,19,20,21,22,23]. In some cases, mutations in specific RBPs are the cause of neurodegenerative diseases, including amyotrophic lateral sclerosis (ALS), which can be caused by mutations in several RBPs, including TIA1 cytotoxic granule-associated RNA binding protein (TIA1), FUS RNA binding protein (FUS), TAR DNA binding protein (TARDBP/TPD-43), and others [24,25,26,27,28], spinocerebellar ataxia type 2 (SCA2), which is caused by a triplet expansion in Ataxin 2 (ATXN2) gene [29,30,31], and Fronto-temporal dementia (FTD), which is caused by mutations in TARDBP [32,33]. In other cases, RBP malfunction is caused by the presence of protein aggregates [19,20], the sequestering of RBP in granules [22], or by the fact that they regulate a different set of target RNAs [16].

## 3. RNA Binding Proteins Have Altered Functions in AD

In AD, RNA processing alterations arise due to changes in the expression, location, and relative abundance of the isoforms of the RBPs expressed, and/or by changes in the sequence (mutations) or expression of the genes they regulate. To assess these changes, both high-throughput transcriptomic and proteomic methods have been used [34,35,36,37,38,39,40,41,42,43,44,45]. However, the large amount of data that these methods generate usually requires the use of additional computational methodologies to summarize the results and identify biologically meaningful modules of co-expressed genes or proteins. Using such approaches, several research studies have identified RBP- or RNA-related modules altered in AD. However, as mRNA expression levels can only explain about 40% of the variance in protein expression in mammalian cells [46,47], transcriptomic and proteomic analyses provide complementary results to assess molecular changes in AD. 

### 3.1. Proteomics Studies Identify RNA Processing Modules Altered in AD

Taking advantage of high-throughput proteomic methods, such as label-free quantification (LFQ) by liquid chromatography tandem mass spectrometry (LC-MS/MS) and related approaches, several works have identified changes in abundance of RBPs [34,35,38,39,48]. Using computational methods, such as weighted correlation network analysis (WGCNA), these studies found that RBPs can be grouped into co-expression modules related to RNA processing, splicing, translation, and gene expression regulation. In these modules, we find RBPs previously known to be found in RNA granules [49,50], as well as other RBPs with low complexity (LC) domains that aggregate in AD, such as small nuclear ribonucleoprotein U1 subunit 70 (SNRNP70/U1-70K) [51], RS repeat-containing proteins, and other spliceosome components [35,38]. Alterations in the expression of these RBP modules are observed across several brain regions, including the dorsolateral prefrontal cortex (DLPFC) [34,35,39], the temporal cortex (TC) [39], and the precuneus (PC), suggesting that these alterations are widespread.

These analyses also revealed several interesting associations between RBP modules and AD. First, it has been established that RBP alterations are not associated with particular cell types, as these modules do not contain specific markers of typical brain cell types such as neurons, oligodendrocytes, astrocytes, microglia, or endothelial cells. Second, it has been demonstrated that RBP module alterations correlate with the disease state of AD patients, as measured by the accumulation of amyloid plaques (CERAD score) and NFT (Braak stages). These changes are also usually significant or show a similar trend in asymptomatic AD patients, suggesting that RBP alterations could already play an important role in the early preclinical phases of AD [34,35,39]. Finally, it has also been shown that alterations in the expression of these RBP modules are independent of apolipoprotein E (APOE) genotype and patient age [39]. Together, these results demonstrate that RBP dysregulation occurs early in the preclinical phases of AD and is exacerbated with disease progression across different brain regions independently of other AD risk factors, such as age and APOE genotype. Thus, these results suggest that RBP alterations could be an independent risk factor of AD. 

Previous studies have shown that alterations in RBP function are common to several neurodegenerative diseases [15,16,17,18,19,20,21]. Thus, a relevant question is whether all these RBP module alterations seen across different brain regions are specific to AD or are shared by other neurodegenerative diseases. To address this question, Johnson et al. [39] compared the expression of protein AD modules in other neurodegenerative diseases. The results from their analyses show that RBP module alterations seen in AD are also seen in frontotemporal lobar degeneration with TDP-43 pathology—a subtype of FTD that, contrary to other FTD subtypes, is characterized by the presence of TARDBP inclusions and the absence of Tau pathology [32]— and corticobasal degeneration (CBD), but not in other neurodegenerative diseases such as ALS, progressive supranuclear palsy, multiple systems atrophy, or Parkinson’s disease (PD). These results suggest that, while RNA processing defects may exist across different neurodegenerative diseases, the specific RBP expression alterations seen in AD are only found in a subset of neurodegenerative diseases.

### 3.2. Transcriptomic Studies Identify Altered Expression of Splicing Factors in AD

mRNA expression levels can only partially explain protein expression levels [46,47]. Therefore, transcriptomic studies provide independent and complementary evidence about the molecular alterations found in AD. Several high-throughput transcriptomics studies have identified alterations in the expression profiles of spliceosome components and other splicing factors, both in the cortex and the hippocampus of AD patients [36,37,40,41,42,43,44,45]. In many cases, the same RBPs and/or functional modules show altered expression profiles, both in neurodegenerative diseases and in aging, although the magnitude of these changes in disease is stronger than in normal aging [36,41]. This is the case for the splicing factors nova alternative splicing regulator 1 (NOVA1), RNA binding fox-1 homolog 1/2 (RBFOX1/2), and KH RNA binding domain containing signal transduction-associated 3 (KHDRBS3); although these are downregulated in both neurodegenerative diseases and aging, they are less expressed in the brains of patients with neurodegenerative diseases than in those from older healthy individuals [41]. Functional analysis of differential gene expression data has allowed us to identify RBPs involved in RNA processing that display altered expression profiles [40,43,52]. More sophisticated analyses combining high-throughput RNA-seq data or microarrays with protein–protein interaction networks [36,37,52], or using RNA-seq data to build gene networks [44], have shown that differentially expressed genes cluster in functional modules related to gene expression regulation, splicing, and RNA processing [36,37,44,52] that are not associated with particular cell types [36]. These findings support the observations from proteomics studies [34,35,37]. Yet, in contrast to what we observed with proteomics data [34,35,39], the overall expression of these genes is downregulated in AD. Among these, we find RBPs known to regulate splicing, such as the polypyrimidine tract binding protein 1 (PTBP1), the serrate RNA effector molecule (SRRT), the small nuclear ribonucleoprotein polypeptide A’ (SNRPA1), and the U2 small nuclear RNA auxiliary factor 1 (U2AF1); RBPs belonging to the polyadenylation complex, such as the cleavage and stimulation factor subunit 2 (CSTF2) and the nudix hydrolase 21 (NUDT21); and RBPs involved in mRNA decay, such as the CCR4-NOT transcription complex subunits 7 (CNOT7) and 9 (CNOT9). Together, these studies show downregulation of RBPs involved in the regulation of splicing, polyadenylation, and mRNA decay. Likewise, these results suggest that these alterations are related to the transcriptomic changes seen in AD.

### 3.3. Single-Cell Transcriptomics Confirm Cell-Type Independent RBP Alterations

In the last 10 years, there has been an explosion of new high-throughput sequencing technologies that allow measuring the expression of genes in individual cells. These technologies, collectively called single-cell transcriptomics of single-cell RNAseq (scRNA-seq), have made possible the study of gene regulation at a new level and characterize the cell composition of tissues and organs such as the brain, to understand not only how gene expression and cell composition change during development and in adulthood [53,54,55,56,57,58], but also how it occurs in neurodegenerative diseases such as AD [59,60,61,62,63,64,65].

Both proteomic and transcriptomic studies have shown that RBP functional modules are not enriched for markers of specific cell types, suggesting that RBP alterations exist across several cell types [34,35,36,39]. To assess this observation, we took advantage of existing single-nuclei RNA-seq (snRNA-seq) datasets [60,64,65] to check whether RBP expression changes in AD were cell-type specific or common across cell types (Montserrat and Plass, in preparation). This analysis identified 59 RBPs differentially expressed in AD compared to control samples (FDR < 0.05; absolute log2 fold change (log2FC) > 0.5). Most of the differentially expressed RBPs were found in neurons, although we also identified RBPs with significantly differential expression profiles in individual clusters of all identified cell types (Figure 2). Among these, only a handful of RBPs were altered across several cell types, suggesting that, while RBPs involved in similar functions were altered across cell types, the specific proteins differentially expressed were mostly cell-type specific. 

### 3.4. RNA Binding Proteins Aggregate in AD

RBPs can reversibly aggregate to form RNA granules, usually through the interaction between LC domains [50,66] and/or basic-acidic (BAD) domains [51]. Under stress conditions, these granules consolidate and are given the name stress granules (SG) [49,67]. Accumulation of SG within cells can lead to the formation of toxic aggregates and thus contribute to the pathology of several neurodegenerative diseases, including AD [22,27,51,68].

Undoubtedly, the most iconic RBP that aggregates in AD is tau protein. Tau aggregates in the brains of AD patients and forms NFTs, which have been associated with progression of the disease [69]. Although initially thought to be involved in the stabilization of microtubule dynamics [70,71], tau is now considered a multifunctional protein [72] that can bind to RNAs [73]. In vitro experiments have shown that tau preferentially binds transfer RNAs (tRNAs) [74] and ribosomal RNAs (rRNAs) [75], and it can affect several regulatory RNA processing steps such transcription, polyadenylation, splicing, and translation [38,75,76,77]. In vivo, tau aggregates located both in the nucleus and the cytoplasm contain small nuclear RNAs (snRNA), small nucleolar RNAs (snoRNAs), and Alu RNAs [78].

Several other RBPs have been found to aggregate in AD (Table 1). Initial work showed RBPs involved in phase transitions, including granule formation, such as TIA1, ZFP36 ring finger protein (ZFP36), and G3BP stress granule assembly factor 1 (G3BP1) aggregate in tautopathy animal models and in AD [79]. This work showed, for the first time, the association of SG with tau pathology in AD. Other than SG formation, these proteins have multiple functions in RNA processing, and regulate other steps such as splicing (TIA1, [80]), RNA stability (ZFP36, [81]), and translation (G3BP1, [82]) among others, which further suggests a connection between protein aggregates and RNA processing defects. Later, it was demonstrated that the formation of SG granules is promoted by tau protein. The presence of tau aggregates promotes the formation of SG, as it increases somatodendritic localization of TIA1, one of the main components of SG, and favors the formation of SG, which are also larger. In turn, the presence of SG favors the aggregation of tau, worsening tau pathology and favoring SG accumulation. This feedback loop not only affects TIA1 but also many of its interactors [83].

To unbiasedly analyze the contents of protein aggregates in AD, several studies have used anionic detergents such as sarkosyl to identify insoluble proteins and characterize them using mass spectrometry methods such as LS-MS/MS. These studies have shown that many RBPs aggregate or have higher aggregation levels in AD, and that aggregation levels increase with disease status [51,87,88,101,103]. Among these proteins, there are many splicing factors such as the serine/arginine repetitive matrix 2 (SRRM2), the RNA binding motif protein 15 (RBM15), and the U2 small nuclear RNA auxiliary factor 2 (U2AF2) [51,88], as well as several U1 small nuclear ribonucleoprotein (snRNP) components such as SNRPA and SNRP70 [51,87,88], polyadenylation factors such NUDT21 [88], and translation regulation factors such as the eukaryotic translation initiation factor 3 subunit A (EIF3A) [51,89] and the eukaryotic translation initiation factor 2 subunit alpha (EIF2S1) [101]. Interestingly, U1 snRNP components already aggregate in asymptomatic AD cases, i.e., in the brains of patients with Aβ deposition in the absence of tau deposition and cognitive impairment [88], reinforcing the idea that RBP alterations may be linked to the development of AD.

As mentioned before, RBP aggregates are often associated with tau protein or NFTs and are less frequently associated with Aβ plaques, although this association is protein-specific (Table 1). For instance, FUS and G3BP1 do not colocalize with phosphorylated tau in frontal cortex brain samples. In contrast, TIA1 and ZFP36 bind phosphorylated tau, and their binding increases with disease severity [79]. 

In contrast, the presence of insoluble snRNPs correlates strongly with both amyloid and tau pathology [79,88,101,103,104]. Together, these studies show that RBP aggregation is common in AD and may be a cause of the altered function of these RBPs in AD, which could partly explain the RNA processing defects described in AD [38,41,43,45,105,106].

## 4. mRNA Changes in AD Are Due to Specific SNPs and Alterations in the Function of the RNA Processing Machinery

There are three main post-transcriptional regulatory mechanisms that impact the selection of gene isoforms expressed in a cell: alternative splicing (AS) [107], alternative polyadenylation (APA) [108], and RNA decay [109]. These mechanisms are mainly regulated by RBPs and define the exonic composition (AS), the RNA 3′end site (APA), and the half-life of the RNAs (RNA decay). Although depicted as independent regulatory processes, several, if not all, of these mechanisms can act on the same gene to define the isoforms that it will express. Moreover, it has to be noted that RBPs are multifaceted proteins that can be involved in the regulation of several of these mechanisms, highlighting the links between the different post-transcriptional regulatory processes in the definition of the transcript repertoire of a cell.

High-throughput omics methods have identified significant changes in the transcriptome and proteome in AD. These changes not only affect the repertoire of genes expressed in AD compared to control samples, i.e., the presence of differentially expressed genes [36,37,40,41,42,44,52,110,111], but also the relative abundance of the specific isoforms that they express [40,41,43,45,105,112,113,114,115], which results in changes in the protein repertoire expressed in AD [35,115]. 

The cause of the changes in isoform usage is not fully understood. In some cases, changes in the expression of RBPs that regulate RNA processing, such as PTBP1, NOVA, and the heterogeneous nuclear ribonucleoprotein C (HNRNPC), have been linked to changes in the inclusion of exons in AD [41,45]. Genome-wide and transcriptome-wide association studies (GWAS and TWAS, respectively) have found clear associations between the presence of single nucleotide polymorphisms (SNPs), which can affect the binding of RBPs, and changes in AS patterns [45,115], particularly among genes known to be associated with AD. However, other works have not found such associations but, rather, a general dysregulation of the splicing machinery that affects the inclusion of multiple exons across genes in the same direction [113]. Taken together, current evidence indicates that both specific genetic variations as well as changes in the expression of RBPs and other unknown factors are responsible for the AS changes seen in AD. Interestingly, some of these changes are also seen in normal aging, although in a milder form [41]. This observation suggests that the mechanisms responsible for such changes are altered in normal aging, although these alterations are increased in AD pathology.

### mRNA Isoform Changes Are Common in Known AD-Risk Genes 

GWAS and TWAS analyses have identified around 50 disease-associated loci in AD, both with familial early onset as well as late onset AD [45,116,117,118,119,120]. Many of these genes present disease-associated changes in their isoforms due to changes in AS and/or APA of their transcripts, including presenilin 1 and 2 (*PSEN1* and *PSEN2*), microtubule-associated protein Tau (*MAPT*), amyloid beta precursor protein (*APP*), phosphofructokinase (*PFKP*), phosphatidylinositol binding clathrin assembly protein (*PICALM),* clusterin (*CLU*), protein tyrosine kinase 2 beta *(PTK2B),* and NDRG family member 2 *(NDRG2)* [35,45,121,122]. Below, we describe in more detail a few examples highlighting how alternative processing of these genes is related to AD.

One example is presenilin genes *PSEN1* and *PSEN2*. These genes form the catalytic domain of the gamma secretase complex, which is responsible for cleavage of APP [123]. Mutations in both genes are associated with familial AD [124,125,126,127,128]. In *PSEN1*, at least four mutations have been described that produce aberrant splicing patterns [124,125,126,127]. While some of these changes lead to reduced protein and/or RNA levels, others directly lead to an increase in the production of Aβ-42 peptide [125,126], which is the main component of amyloid plaques [129]. 

Splicing variants from *APP* are also directly associated with AD. *APP* can generate multiple isoforms through alternative splicing of exons 7, 8, and 15 [130]. The expression of the longer *APP* isoforms, APP751 and APP770, is increased in the brains of AD patients, and it is associated with higher Aβ deposition [131]. Relative expression of *APP* long isoforms is regulated by miRNA-124, as well as by the expression levels of PTBP1 and PTBP2 proteins [132], suggesting the involvement of these regulators in AD.

MAPT is also regulated by AS. *MAPT* generates multiple splicing isoforms that are developmentally and spatially regulated [133]. Alternative splicing of exon 10, which affects the balance of 3R and 4R tau isoforms, and thus tau aggregation, is influenced by mutations located around the 5′ splice site, which leads to exon skipping, as well as by the function of specific RBPs such as FUS, the serine- and arginine-rich splicing factor 2 (SRSF2), and TARDBP [133,134]. Interestingly, these proteins are known to be dysregulated [79,135] and/or aggregate in AD [79]. Therefore, these results suggest a clear connection between the presence of RBP aggregates and tau pathology in AD. On the other hand, a new truncated *MAPT* protein isoform that does not aggregate has recently been discovered. This new isoform, which is generated through an intronic polyA site in intron 12, is lowly expressed in AD patients [136]. Together, these pieces of evidence highlight the complex relation that exist between RBPs, AS, APA and tau-driven neurodegeneration in AD.

## 5. miRNAs in AD

miRNAs are short noncoding RNAs (21–24 nt) implicated in many cellular processes, including proliferation, differentiation, senescence, stress response, and apoptosis [137,138,139,140]. Initially, miRNAs are transcribed as long primary pri-miRNA transcripts and then processed into precursor pre-miRNAs by the microprocessor complex, which consists of type-III Rnase Drosha and DiGeorge critical region 8 (DGCR8) [141,142]. Pre-miRNAs are then exported to the cytoplasm by exportin 5 (XPO5), where they are cleaved by dicer to mature miRNAs [143,144,145]. Usually, one strand of the mature miRNA duplex is loaded into the RNA-induced silencing complex (RISC) to suppress the stability and/or translation of their target mRNAs through partial complementarity to the 3′ untranslated region (3′UTR) of their targets [146,147,148] (Figure 1). However, in some cases, miRNAs have alternative functions and instead of downregulate mRNA expression, they promote gene expression. For example, miR-346 stimulates the translation of APP indirectly by interacting with its 5′UTR. This interaction prevents the recruitment of a translation suppressor, aconitase 1 (ACO1), and thus indirectly stimulates APP translation [149]. In a non-canonical manner, miRNAs can also act as signaling molecules, as reported for miR-let-7b, which activates toll-like receptors (TLRs) [150], or can interact with ncRNAs, such as lncRNAs or circRNAs. Some of these interactions will be described in the next sections of this review. Intriguingly, the primary transcripts of some miRNAs can even encode peptides called miPEPs (miRNA-encoded peptides), such as pri-miR-171b and pri-miR-165a [151,152]. 

miRNAs are abundant in the central nervous system and their dysregulation is closely related to human brain disorders. Dicer depletion in the adult forebrain causes a mixed neurodegenerative phenotype [153], indicating that miRNAs are engaged in regulation of neurodegeneration-associated pathways. Moreover, numerous studies reported miRNA contribution to AD-related pathologies, including the formation of Aβ aggregates and NFTs, and the induction of tau phosphorylation. In the following sections, we provide an overview of miRNAs’ association with AD pathologies and their potential value as AD biomarkers.

### 5.1. miRNAs Regulate APP Expression

Accumulating evidence suggests that miRNAs can have an important function in the regulation of APP-related neuropathological conditions. miRNAs can function directly by regulating the expression levels of APP or modulating the expression of different splicing isoforms of APP gene [132]. Patel et al. [154] were the first to demonstrate miRNAs’ involvement in the regulation of APP expression. Using a reporter construct carrying APP 3′UTR, they showed that miR-106a and miR-520c bind to the 3′UTR of APP and can negatively regulate reporter expression and cause translational repression, significantly reducing APP protein levels. Following studies have also shown that the miR-20 family (miR-20a, miR-17-5p, and miR-106b) [155], miR-101 [156], and miR-16 [157] target APP 3′UTR and downregulate APP protein levels. In vivo inhibition of endogenous miR-101 using miR-101 sponge (pLSyn-miR-101 sponge) resulted in cognitive impairment in a mouse model, which is associated with increased hippocampal expression of APP and overproduction of amyloid beta Aβ-42 [158]. 

Another miRNA implicated in APP expression regulation is miR-153. miR-153 delivery in both HeLa cells and primary human fetal brain cultures significantly reduced APP expression, while delivery of a miR-153 antisense inhibitor resulted in significantly elevated APP expression. miR-153 levels are reduced in a subset of human AD brain specimens with moderate AD pathology, which presented elevated APP levels. These results suggest that miR-153 may have relevance to AD etiology, and low miR-153 levels may cause increased APP expression in AD patients [159].

Several miRNAs have also been implicated in the regulation of the beta-secretase BACE1, which is an essential enzyme for the generation of Aβ. For example, miR-29a/b-1 represses BACE1 expression both in vitro and in vivo. In turn, decreased levels of miR-29a/b-1 induce production of Aβ [160]. miR-29c is downregulated in sporadic AD brains and was associated with abnormally high levels of BACE1. In vitro overexpression experiments of miR-29 in SHSY5Y cells demonstrate the specificity of miR-29-BACE1 interaction [161]. miR-107 also regulates BACE1 expression. An early decrease in miR-107 expression may accelerate AD progression [162,163]. Furthermore, miR-107 was shown to prevent neurotoxicity and blood–brain barrier dysfunction induced by Aβ [164,165]. Other negative regulators of BACE1 include miR-298/328 [166], miR-195 [167], miR-135a [168], miR-135b [169], miR-9 [170], and miR-298 [171], indicating the complexity of the miRNA network associated with BACE1 regulation.

### 5.2. miRNA Function in Aβ Clearance

The expression of several miRNAs is associated with Aβ clearance. In sporadic AD patients, upregulation of miR-128 results in impaired clearance of Aβ. miR-128 targets several lysosomal enzymes, including cathepsin B, D, S, β-Galactosidase, α-Mannosidase, and β-Hexosaminidase; thus, its upregulation impairs the lysosomal system [172]. miR-34a is also implicated in the regulation of Aβ clearance by inhibiting the expression of triggering receptor expressed on myeloid cells 2 (TREM2) [173,174]. TREM2 is a microglial cell surface receptor implicated in recognizing and digesting Aβ and extracellular amyloidogenic debris, and it plays a crucial role in Aβ clearance [175].

Another miRNA, miR-1908, also plays an important role in inhibiting Aβ clearance through repressing APOE expression at the mRNA and protein in cell lines. In AD patients, miR-1908 and APOE showed a reciprocal expression pattern, implicating miR-1908 in the APOE regulatory loop in Aβ clearance [176]. 

Several miRNAs have also been implicated in regulation of the ubiquitin–proteasome system, which is relevant for Aβ and tau clearance. For example, the regulatory loop formed by miR-7 and the circRNA CDR1 antisense RNA (ciRS-7/CDR1as) has been linked to the depletion of ubiquitin conjugating enzyme E2A (UBE2A) in AD patients [177]. miR-9 was also shown to be involved in regulation of the ubiquitination factor E4B (UBE4B)—the ubiquitin enzyme mediating tau degradation [178].

### 5.3. miRNAs and Tau-Related Pathologies

Several studies suggested that various miRNAs can regulate different steps of tau processing, including splicing and posttranslational modifications. First of all, the observation that dicer depletion in adult forebrain is accompanied by hyperphosphorylation of endogenous tau indicates a direct role of miRNA in tau-related neuropathies. This was later shown to be directly linked to the downregulation of miR-15 family members in AD patient brains. miR-15 members downregulate the mitogen activated protein kinases 1 and 3 (MAPK1/3). MAPK1/3 are known to phosphorylate tau, which has been associated with the presence of NFTs and senile plaques. miR-26a instead regulates another protein kinase, the glycogen synthase kinase 3 beta (GSK3B), which is also known to hyperphosphorylate tau and is associated with both Aβ generation and NFT formation in AD brains [179]. Moreover, tau phosphorylation is regulated by miR-124-3p. miR-124-3p inhibits the translation of CAPN1 mRNA and the subsequent conversion of p35 to p25 and the formation of the p25/CDK5 complex, which controls the abnormal tau phosphorylation [180].

As previously mentioned, several brain miRNAs, including miR-124, miR-9, miR-132, and miR-137, affect the splicing of MAPT exon 10, creating an imbalance between 3R and 4R tau isoforms and leading to taupathies [132,181]. miRNAs exert this effect by changing the expression of the splicing factors PTBP1 and PTBP2, which regulate exon 10 inclusion [182]. 

miRNAs also regulate tau acetylation, which is closely associated with abnormal tau protein aggregation and AD progression. Several miRNAs, including miR-9, miR-212, and miR-181c, inhibit the expression of sirtrulin 1 (SIRT1), which deacetylates tau. Downregulation of SIRT1 in AD brains correlates with a significant accumulation of phosphorylated tau [183]. 

Taken together, miRNAs have been characterized as important players in the regulation of APP and tau in AD and other tau-related pathologies. In recent years, they also gained considerable attention as potential biomarkers for AD [184,185,186,187,188]. There have been several attempts to profile miRNA expression changes in brain or circulating fluids of AD patients in order to elucidate miRNA association with disease progression [189,190]. Although miRNAs can be reproducibly measured in serum and CSF without pre-amplification [191], their expression levels do not always correlate across body fluids. Therefore, unification of analytical protocols, storage times, and quantification methods is necessary to define consistent miRNA biomarkers for AD [191].

## 6. Circular RNA

circRNAs are a newly discovered class of stable, covalently closed, naturally occurring RNAs, with widespread expression in eukaryotic cells [192,193,194,195]. They are produced by the spliceosome in noncanonical back-splicing reactions [196,197] (Figure 1). Their extraordinary stability, due to their resistance to exonucleolytic activity, allows them to escape classical RNA turnover, and offers the ability to function differently than normal, linear RNAs, such as mRNAs or lncRNAs. The majority of circRNAs derive from exons of protein coding genes and, although thousands of these have been identified and many exist as predominant transcript isoforms, little is known about their function and association with human diseases.

Their exceptionally high abundance in the brain, particularly in synapses, and their activity-dependent expression makes them new potential players in synapse dysfunction disorders. Until recently only a few circRNAs have been characterized and associated with AD. circRNAs have been shown to be involved in regulation of several AD-related pathologies, such as neuroinflammation, Aβ-accumulation, and oxidative stress (Table 2). In familial AD, Aβ peptides are generated from the full-length APP protein via dysregulated proteolytic processing. A circRNA termed circAβ-a is produced from APP-locus and has been shown to be efficiently translated into a novel Aβ-containing Aβ175 polypeptide (19.2 KDa) that can be processed into Aβ peptides in both cultured cells and human brain. This indicates that circRNAs can be involved in an alternative pathway of Aβ biogenesis [198].

### circRNAs’ Function in AD

The biochemical heterogeneity and wide expression range of circRNAs suggest potential functions, such as delivery vehicles, RBP sponges, assembly of RBP factories, or as potential templates for translation [192,205]. However, the best characterized function of circRNAs is associated with miRNA regulation. circRNAs act as natural miRNA antagonists, i.e., “sponges”. For example, ciRS-7/CDR1as, antisense to cerebellar degeneration-related protein 1 (CDR1), is densely bound by miRNA effector complex and harbors 63 conserved miR-7 binding sites [192,206]. ciRS-7/CDR1as has also been reported to play a crucial role in the pathogenesis of AD. It has been shown that ciRS-7/CDR1as is downregulated in the brains of AD patients [177,199] and affects the expression of UBE2A, which is crucial for clearance of Aβ via proteolysis in AD [177]. ciRS-7/CDR1as also plays an essential role in regulating protein levels of APP and BACE1 [207]. Interestingly, overexpression experiments of ciRS-7/CDR1as indicate that it may have a neuroprotective function by regulating expression of ubiquitin carboxyl-terminal hydrolase L1 (UCHL1), which reduces Aβ production via the proteasomal and lysosomal degradation of APP and BACE1 [207].

Apart from ciRS-7/CDR1as, additional studies have identified circRNAs dysregulated in cortical areas in AD [204]. Among them, two circRNAs, circHOMER1 and circCORO1C, are significantly correlated with AD neuropathology status. Interestingly, circHOMER1 has multiple binding sites for miR-651, and circCORO1C binds to miR-105, which was predicted to target both APP and SNCA42 and to be associated with AD pathology [204].

Several works have focused on the characterization of circRNA–miRNA interaction networks that are dysregulated in AD. Zhang et al. built a circRNA–ceRNA network (ADcirCeNET) and used it to identify candidate circRNAs that could function as miRNA sponges. This work identified circRNA KIAA1586 as a miRNA sponge for several miRNAs, including miR-29b, miR-101, and miR-15a, which might regulate different AD-associated genes [208]. Some of the interactions predicted by ADcirCeNET have been previously validated using AD cell models [209,210]. For instance, the overexpression of miR-29b suppressed the mRNA and protein expression of BACE1 and reduced the Aβ42 level. Other examples are miR-101, which can bind to the 3′UTR of APP to reduce the level of APP in AD, and miR-15a, which is significantly dysregulated in sporadic AD patients [160], and is predicted to bind to and regulate BACE1 and APP. 

Another circRNA that functions as a miRNA antagonist in AD context is circHDAC9. circHDAC9 functions as a miR-138 sponge and has a decreased expression in the serum of AD patients [201]. The downregulation of ADAM Metallopeptidase Domain 10 (ADAM10) by miR-138 promotes Aβ production, while the expression of circHDAC9 has the opposite effect and significantly suppresses Aβ peptide production in vitro. In turn, overexpression of miR-138 target SIRT1 reduces miRNA-induced inhibition of ADAM10 and Aβ accumulation in vitro [201]. All these results suggest that circHDAC9 could be used as a therapeutic target, as increased expression of circHDAC9 could prevent Aβ accumulation.

Using 2D models, Yang et al. reported yet another circRNA–miRNA regulatory function implicated in AD-related neuroinflammation [200]. CircRNA_0000950, by acting as a miR-103 sponge, leads to upregulation of the prostaglandin-endoperoxide synthase 2 (PTGS2) and inflammatory cytokines, such as interleukin 6 and 1β (IL6 and IL1B), and the Tumor Necrosis Factor (TNF), causing an increase of neuronal cell apoptosis and suppression of neurite outgrowth [200]. 

Several circRNAs have also been identified as being dysregulated in mouse models of AD. For instance, in senescence-accelerated mouse-prone 8 (SAMP8) mice, circNF1–419 has been associated with early neuropathological changes related to autophagy in AD. circNF1–419 interacts with dynamin-1 and adaptor protein 2 B1 (AP2B1), regulating their mRNA splicing, stabilization, and translation, and its overexpression reduces AD marker proteins (Tau, p-Tau, Aβ1–42, and APOE), as well as aging and inflammatory factors such as TNF and the nuclear factor kappa B subunit 1 (NKFB1), indicating delayed senile dementia and progression of AD [211]. Another example is circRNA_017963, which is downregulated in 10-month-old SAMP8 mice. GO pathway analysis suggests that circRNA_017963 may play a role in autophagosome assembly, exocytosis, and synaptic vesicle cycle related to AD pathogenesis [212]. In the same model, after Panax notoginseng saponins (PNS) treatment, mmu_circRNA_013636 and mmu_circRNA_012180 were also predicted to be involved in AD-associated biological processes and pathways [213].

Analysis of an APP/PS1 model revealed deregulation of several abundant circRNAs, such as circTulp4. circTulp4 localizes predominantly in the nucleus where it interacts with U1 snRNP and RNA polymerase II, regulating the expression of its parental gene, Tulp4, and might contribute to the development of AD [202].

All studies described above suggest that circRNA may play a critical, yet now well-explored, role in AD. However, only miRNA regulatory function has been the reported for larger sets of circRNA; therefore, other mechanisms of circRNA action in AD require more investigation. Besides, more human expression profiling studies are necessary to define AD-associated circRNAs with potential biomarker or therapeutic target values.

## 7. LncRNAs

LncRNAs are the larger subgroup ncRNAs. LncRNAs are defined as transcripts longer than 200 nucleotides that lack an open reading frame [214]. Most of them are localized in the nucleus and play essential roles in the regulation of gene expression at the transcriptional, post-transcriptional, and translational levels, interacting with DNA, mRNAs, proteins, and miRNAs. Increasing evidence suggests that aberrant expression of lncRNAs correlates with AD progression. LncRNAs have been associated with different aspects of AD pathology, such as regulation of Aβ peptide, tau, inflammation, and neuronal cell death. Here, we discuss the possible roles of lncRNAs in AD pathology and their utility as diagnostic and therapeutic targets.

Initial studies showed that several hundreds of lncRNAs were differentially expressed in AD animal models. In 2015, Lee et al. reported 205 lncRNAs significantly dysregulated in 3xTg-AD compared to control mice [215]. An additional 249 were identified in the hippocampus of APP/PS1 transgenic mice [216]. Analysis of human postmortem tissue samples also identified hundreds of lncRNAs dysregulated in AD [217], whose expression aligns well with the clinical diagnosis of AD [217,218,219].

Several lncRNAs have been studied in detail and have been shown to regulate different aspects of mRNA metabolism related to AD, including regulation of transcription, splicing, mRNA stability, and protein synthesis. We summarize all the characterized lncRNAs in Table 3, and some examples will be described in more detail.

BACE1-AS is the antisense transcript to BACE1, which plays a crucial role in the toxic Aβ production. BACE1-AS regulates BACE1 expression at both the mRNA and protein levels. BACE1-AS expression is elevated in AD brains and its expression correlates with Aβ accumulation [226]. Expression of both BACE1 and BACE1-AS is upregulated by stressors (Aβ, high temperature, serum starvation, high glucose, or staurosporine) and thus enhance APP processing and Aβ production [226]. Downregulation of BACE1-AS in SHSY5Y cells reduces Aβ production by BACE1 and plaque deposition [168], while, in animal models, it results in several behavioral and physiological deficits, including reductions in synaptic plasticity and memory loss. BACE1-AS regulation of BACE1 is associated with the masking of miR-485-5p binding site in 3′UTR of BACE1 [228]. The expression of both BACE1 and BACE1-AS is regulated by the RBP ELAV like RNA binding protein (ELAVL1), which promotes AD-related pathological changes [227].

The brain cytoplasmic RNA 1 (BCYRN1) is a 200 nt lncRNA that localizes to synaptodendritic neuronal compartments, where it modulates local protein synthesis by targeting eukaryotic initiation factor 4A (EIF4A). In normal aging, BCYRN1 expression is reduced by more than 60%, while it is significantly upregulated in AD-involved brain areas and correlates with disease progression [236]. Surprisingly, previous studies have shown that BCYRN1 was downregulated in AD patients compared to control and other dementias [238]. The contradictions in these studies may result from differences in analyzed brain regions or the relevant stages of the disease.

51A is an antisense transcript to the intron 1 of the sortilin-related receptor 1 (SORL1). SORL1 is an endocytic receptor involved in APP trafficking [239,240,241]. Downregulation of SORL1 increases secreted APP production and subsequent Aβ formation. Moreover, 51A regulates SORL1 alternative splicing, resulting in production of alternatively spliced SORL1 variants that affect Aβ formation [224]. 51A was also found to be elevated in in vitro models and in AD patient brains [224].

Massone et al. [222] describe another lncRNA, called 17A, which is 159-nt long and derives from the human G-protein-coupled receptor 51 gene (GPR51, GABA B2 receptor). This lncRNA is upregulated in cerebral tissues derived from AD patients. The overexpression of 17A in SHSY5Y cells results in the synthesis of an AS isoform that interferes with GABA B2 signaling and enhances the secretion of Aβ peptide [222,223].

LRP1-AS is a 1387 nt lncRNA antisense to the LDL receptor-related protein 1 (LRP1) gene. LRP1-AS negatively regulates LRP1 expression at both RNA and protein levels [220]. LRP1-AS directly binds to high-mobility group box 2 (HMGB2) and inhibits its activity to enhance sterol regulatory element binding transcription factor 1 (SREBF1A) dependent transcription of LRP1. The levels of LRP1-AS are significantly increased in AD [242].

## 8. Conclusions

The relationship between RBPs, RNA processing mechanisms, and AD is complex, and we have yet to fully understand it. In this review, we have provided an overview of how RNA dynamics are altered in AD, how they are regulated by RBPs, and what impact they have on the expression of both long and short RNA species. Furthermore, we provide an insight into how the changes in the abundance of specific RNA species are linked to the pathology of AD. In the text, we have highlighted some of the most well-described changes known, which, in many cases, have functional consequences. However, the common use of high-throughputs methods shows that RBP and RNA processing alterations go far beyond what has been studied. Considerable research is required into this topic in order to elucidate the complex function of RNA metabolism in AD.

## Figures and Tables

**Figure 1 molecules-26-05113-f001:**
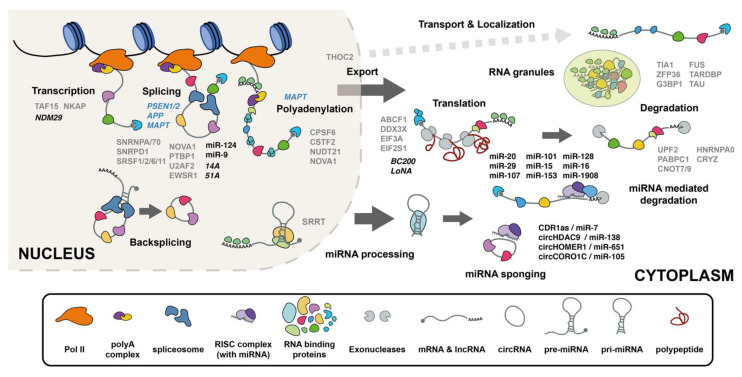
RNA processing in AD. Schematic representation of the main processes regulated by RBPs and RNAs within cells and their alterations in AD. The figure depicts the biogenesis of the main coding and non-coding RNA species, including mRNAs, lncRNAs, miRNAs, and circRNAs, as well as the main RNA regulatory processes, including transcription, splicing, RNA transport, storage, translation, and RNA degradation. Additionally, we highlight some of the known RNA processing alterations described in AD, including (1) changes in the expression of mRNA isoforms of AD-related genes (blue); (2) RBPs with altered function (grey); and (3) miRNA and lncRNA with altered expression (black). Genes, RNAs, and RBPs appear next to the relevant biological process to which they are related.

**Figure 2 molecules-26-05113-f002:**
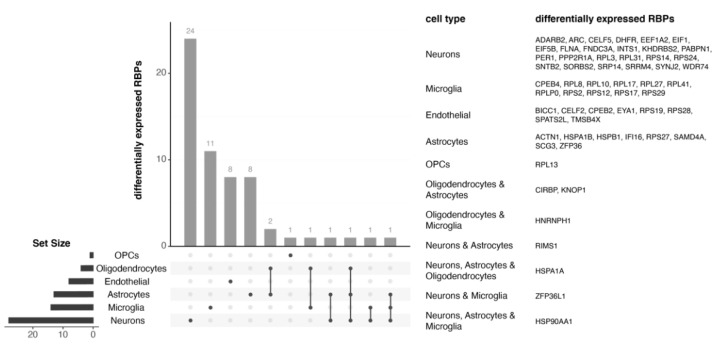
RBPs differentially expressed in individual cell populations in AD. 59 RBPs are differentially expressed at the mRNA level in snRNA-seq data. The majority of them are differentially expressed in a single cell type. Most differentially expressed RBPs are found in neurons (24 RBPs), although we also found a significant number of them in microglia (11 RBPs), astrocytes (8 RBPs), and endothelial cells (8 RBPs).

**Table 1 molecules-26-05113-t001:** RBPs that aggregate in AD.

GENE NAME	Description	Tau Association	Aβ Association	References
MAPT	Microtubule-associated protein tau	YES	YES	[84]
MSI1	Musashi 1	YES	NO	[85,86]
MSI2	Musashi 2	YES	NO	[85,86]
SNRNP70	U1 small nuclear ribonucleoprotein 70 kDa	YES	NO	[51,87,88]
LUC7L3	LUC7 like 3 pre-mRNA splicing factor	YES	NO	[38,51]
SRRM2	serine/arginine repetitive matrix 2	YES	YES	[51,87,88]
PRPF40A	pre-mRNA processing factor 40 homolog A	NO	NO	[51]
	ADP ribosylation factor like GTPase 6 interacting protein 4	NO	NO	[51]
THOC2	THO complex 2	NO	NO	[51]
PRPF4B	pre-mRNA processing factor 4B	NO	NO	[51]
RNF20	ring finger protein 20	NO	NO	[51]
RBM15	RNA binding motif protein 15	NO	NO	[51]
NKAP	NFKB-activating protein	NO	NO	[51]
RNPS1	RNA binding protein with serine-rich domain 1	NO	NO	[51]
ACIN1	apoptotic chromatin condensation inducer 1	NO	NO	[51]
GPATCH8	G-patch domain containing 8	NO	NO	[51]
EIF3A	eukaryotic translation initiation factor 3 subunit A	YES	NO	[51,89]
YTHDC1	YTH domain containing 1	NO	NO	[51]
SRRT	Serrate RNA effector molecule	YES	YES	[51,89]
SRSF11	Serine- and arginine-rich splicing factor 11	NO	NO	[51]
PLEC	Plectin	YES	NO	[51,90]
TCERG1	transcription elongation regulator 1	YES	NO	[51,90]
PRPF38B	pre-mRNA processing factor 38B	NO	NO	[51]
	Zinc finger protein 638	NO	NO	[51]
CPSF6	Cleavage- and polyadenylation-specific factor 6	NO	NO	[51]
SNRPA	U1 small nuclear RNP-specific A	YES	NO	[38,87,88]
SYNJ1	Synaptojanin 1	YES	NO	[87,91]
RIMS1	Regulating synaptic membrane exocytosis 1	NO	NO	[87]
DDX46	DEAD-Box Helicase 46	NO	NO	[87]
TARDBP	TAR DNA-binding protein 43	YES	YES	[79,92,93,94,95]
TIA1	T-cell intracellular antigen 1	YES	NO	[79]
G3BP1	Ras GTPase-activating protein-binding protein 1	NO	NO	[79]
ZFP36	ZFP36 Ring Finger Protein	YES	NO	[79]
FUS	Fused in sarcoma	NO	NO	[79]
SOD1	Superoxide dismutase	YES	YES	[96,97]
RBM45	RNA binding motif protein 45	NO	NO	[98]
SRSF6	Serine- and arginine-rich splicing factor 6	YES	YES	[89]
SRRM1	serine/arginine repetitive matrix 1	YES	NO	[89]
MAK16	MAK16 homolog	NO	YES	[89]
ABCF1	ATP binding cassette subfamily F member 1	YES	NO	[89]
SRSF1	Serine- and arginine-rich splicing factor 1	YES	NO	[89]
DDX3X	DEAD-Box helicase 3 X-linked	YES	NO	[89]
UTP20	UTP20 small subunit processome component	YES	NO	[89]
SNRPD1	Small nuclear ribonucleoprotein Sm D1	YES	NO	[88]
SLIRP	SRA stem-loop-interacting RNA-binding protein, mitochondrial	YES	NO	[88,99]
U2AF2	Splicing factor U2AF 65 kDa subunit	NO	NO	[88]
UPF2	Regulator of nonsense transcripts 2	NO	NO	[88]
NCL	Nucleolin	YES	NO	[88,90,100]
NUDT21	Cleavage and polyadenylation specificity factor subunit 5	YES	YES	[88]
CRYZ	Quinone oxidoreductase	NO	NO	[88]
DARS2	Aspartate tRNA ligase, mitochondrial	NO	NO	[88]
EWSR1	EWS RNA binding protein 1	YES	NO	[101]
TAF15	TATA-Box binding protein-associated factor 15	YES	NO	[101]
RPL7	Ribosomal protein L7	YES	NO	[101]
DDX5	DEAD-Box helicase 5	YES	NO	[101]
HNRNPA0	Heterogeneous nuclear ribonucleoprotein A0	YES	NO	[101]
PABPC1	Poly(A) binding protein cytoplasmic 1	YES	NO	[101]
DDX6	DEAD-Box helicase 6	YES	NO	[101]
EIF2S1	Eukaryotic Translation Initiation Factor 2 Subunit Alpha	YES	NO	[101]
SFPQ	Splicing factor proline, glutamine rich	YES	NO	[48,102]

**Table 2 molecules-26-05113-t002:** circRNAs with known function in AD.

circRNA	Function in AD	References
ciRS-7/ciRS-7/CDR1as	regulating protein levels of APP and β-secretase (BACE1)	[177,199]
CircRNA_0000950	miR-103, function in apoptosis, neurite outgrowth, and neuroinflammation in AD	[200]
circHDAC9	miR-138 sponge regulation of ADAM10 and Aβ production	[201]
circTulp4	interacts with U1 snRNP and RNA polymerase II to modulate the transcription of its parental gene, Tulp4	[202]
circHOMER1	binding sites for mir miR-651	[203]
circCORO1C	bind to miR-105, which is predicted to target both APP and SNCA42	[204]
circRNA KIAA1586	sponge for several miRNAs including miR-29b, miR-101, and miR-15a	[203]

**Table 3 molecules-26-05113-t003:** Functional lncRNAs in AD.

Process Regulated	circRNA	Function in AD	References
Transcription	*LRP1-AS*	Inhibits the activity of Hmgb2 to enhance Srebp1a-dependent transcription of Lrp1	[220]
	*NDM29*	Upregulated in AD. Enhances ratio of Aβ42/Aβ40 and accumulation of Aβ	[221]
Splicing	*17A*	Synthesis of an alternative splicing isoform that interferes with GABA B2 signaling and enhances the secretion of amyloid β peptide (Aβ)	[222,223]
	*51A*	Regulates SORL1 gene splicing	[224]
RNA stability	*EBF3-AS*	Upregulated in AD mouse models Regulates EBF3 stability and promotes neuronal apoptosis	[225]
	*BACE1-AS*	Upregulated in AD. Regulates BACE1 stability and enhances Aβ production	[226,227,228]
	*SOX2OT*	Dysregulated in mouse AD models. Regulates SOX2 expression	[229,230]
	*NEAT1*	Reduced in AD and in mouse AD models. Inhibits the expression of endocytosis-related genes and decreases Aβ clearance Regulates miR-124/BACE1 axis. Target of miR-107	[231,232,233]
	*NAT-RAD18*	Upregulated in AD. Regulates RAD18	[234]
	*BDNF-AS*	Negatively regulates BDNF expression	[235]
Translation	*BC200*	Dysregulated in AD Modulates local protein synthesis by targeting EIF4A	[236]
	*LoNA*	Upregulated in hippocampus of AD mouse. Regulates synaptic plasticity by regulating ribosomal assembly and protein translation	[237]

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
