# Peer review of "RNA Dynamics in Alzheimer’s Disease"

_molecules, 2021, doi:10.3390/molecules26175113_

Round 1
Reviewer 1 Report
In this work, Rybak-Wolf and Plass have examined the importance of RNA metabolism in Alzheimer disease. Alterations, in RNA metabolism has been recently observed to be a major neuropathological pathway in a variety of neurodegenerative conditions, making this review both timely and interesting. In general, the review is very well written and only some minor modifications/corrections should be considered by the authors:
- In page 2, line 46: the verb in “Therefore, suggest that the…” in not well conjugated. Please change or rephrase.
- In page 4, line 134: “in frontotemporal lobar degeneration with TDP-43 pathology, a subtype of FTD…” please add that the other subtype of FTD is characterized by Tau pathology.
- In page 4, line 157: “that are nor associated…” should be “that are not associated..”
- For the sake of completeness, Figure 2 should contain a list of the 59 RBPs shown to be differentially expressed at the mRNA level in snRNA-seq data. This could either be done by making boxed insets in the figure or adding a separate table that shows the names of these factors.
- In page 5 or 9, when introducing Tau/MAPT, the authors should consider mentioning that several reports have shown the factor TDP-43 as capable of affecting splicing of this gene, thus strengthening the possible connections between all these disease factors.
- In page 13, line 525: “3xTg-AD compare to control mice..” should be compared.
- In page 13-14, the phrase starting at line 528 is not very clear and should also be reworded.
Reviewer 2 Report
This manuscript submitted by Agnieszka Rybak-Wolf and Mireya Plass describe the relationship between the alteration of RNA species and AD in detail. It is comprehensive and informative, showing how expression, function, abundance of differential RNA species affect the pathology of AD. This manuscript is well written in general, but minor revision is needed. I would recommend to publish on Molecules after revision based on the comments below.
- Figure 1 is beautiful and shows many details related to RNA regulation. I wonder if it is possible to add some contents that can indicate their relationship with AD. Alternatively, please consider to summarize the whole review using an extra "figure 3" that illustrates the RNA dynamic alteration and AD if possible.
- Please check the grammar in the sentence in line 75, "can be caused..." and line 77 "cause by".
- Delete the sentence at the beginning of section 3, "This section may be...".
- Please check if all the abbreviations have full description when occur at the first time. For example, APOE is not clearly illustrated.
- It would be better to add references after sentence in line 87, "To access these..."
- The sentence between line 160 and 163 is not well-written and confusing.
- The Y label of Figure 2 is a bit far from the axis.
